# A Deep Learning-Based Framework for Retinal Disease Classification

**DOI:** 10.3390/healthcare11020212

**Published:** 2023-01-10

**Authors:** Amit Choudhary, Savita Ahlawat, Shabana Urooj, Nitish Pathak, Aimé Lay-Ekuakille, Neelam Sharma

**Affiliations:** 1University School of Automation and Robotics, G.G.S. Indraprastha University, New Delhi 110092, India; 2Maharaja Surajmal Institute of Technology, G.G.S. Indraprastha University, New Delhi 110058, India; 3Department of Electrical Engineering, College of Engineering, Princess Nourah Bint Abdulrahman University, P.O. Box 84428, Riyadh 11671, Saudi Arabia; 4Department of Information Technology, Bhagwan Parshuram Institute of Technology (BPIT), G.G.S. Indraprastha University, New Delhi 110078, India; 5Department of Innovation Engineering, University of Salento, 73100 Lecce, Italy; 6Department of Artificial Intelligence and Machine Learning, Maharaja Agrasen Institute of Technology (MAIT), G.G.S. Indraprastha University, New Delhi 110086, India

**Keywords:** artificial intelligence, image processing, diseased state of retina, transfer learning, neural networks, VGG-19 architecture, performance analysis

## Abstract

This study addresses the problem of the automatic detection of disease states of the retina. In order to solve the abovementioned problem, this study develops an artificially intelligent model. The model is based on a customized 19-layer deep convolutional neural network called VGG-19 architecture. The model (VGG-19 architecture) is empowered by transfer learning. The model is designed so that it can learn from a large set of images taken with optical coherence tomography (OCT) and classify them into four conditions of the retina: (1) choroidal neovascularization, (2) drusen, (3) diabetic macular edema, and (4) normal form. The training datasets (taken from publicly available sources) consist of 84,568 instances of OCT retinal images. The datasets exhibit all four classes of retinal disease mentioned above. The proposed model achieved a 99.17% classification accuracy with 0.995 specificities and 0.99 sensitivity, making it better than the existing models. In addition, the proper statistical evaluation is done on the predictions using such performance measures as (1) area under the receiver operating characteristic curve, (2) Cohen’s kappa parameter, and (3) confusion matrix. Experimental results show that the proposed VGG-19 architecture coupled with transfer learning is an effective technique for automatically detecting the disease state of a retina.

## 1. Introduction

Deep learning (DL) for medical image analysis has brought a big improvement in the detection of disease and its diagnosis. In earlier scientific methods, the detection of disease is less reliable, prone to erroneous detection, and even takes a large amount of time for proper conclusions. However, in recent times, the biomedical field started using machine learning techniques on a large scale for disease detection, which provides high accuracy results with concise timing. One of the most common causes, why people lose eyesight at a significantly early age, are diseases or damage to the retina (a fine layer located at the inner backside of the human eyeball) causing adverse effects on the retina. This paper throws light on the computer-assisted automated detection procedures of retinal diseases (namely choroidal neovascularization (CNV), diabetic macular edema (DME), normal, and drusen) using transfer learning. Some OCT images of the retina representing the abovementioned three diseases and one normal state are shown in Figure 1.

Retinal disease detection is a classical classification problem for machine learning. The present work solves the problem by automating the detection of diseases from their corresponding scanned optical coherence tomography (OCT) images of the retina among 4 different classes of retinal states (including CNV, drusen, DME, and normal). As the use of imaging technique, optical coherence tomography (OCT) is increasing day by day in medical science, a computer assisted diagnosis system could be very successful and reliable in the automatic detection of retinal diseases and will help in their treatment and monitoring. In case of conventional methodologies, the process of automated retinal disease diagnosis comprises a large amount of image pre-processing, which will then be fed into a shallow neural network, which is a relatively time-consuming process. Because of these limitations of shallow networks, we will be implementing transfer learning techniques. The motivation of present research is the challenges faced by the researchers in the area of retinal diseases diagnosis.

Machine learning (ML) models and DL models continue to provide significant performance in the diagnosis of retinal disease. In the present work, DL methods are preferred because of their capability to understand and process biological data and ability to extract high-level abstract features from the sample images. The objective of our study is to employ a transfer learning methodology over a VGG-19 network with pre-trained weights of ImageNet dataset. The performance is evaluated and compared with the pre-existing methods for retinal disease detection. The performance and accuracy of the proposed model are validated over an independent testing dataset and on a foreign dataset (unknown dataset to model), named DHU (stands for Duke University, Harvard University, and the University of Michigan) dataset of OCT scanned retinal images of 45 patients. The application of this work is to automate retinal disease detection and provide a computer-assisted diagnosis for many eye conditions like macular edema, age-related macular degeneration, and diabetic retinopathy. The key contributions of the proposed work are as follows:The proposed work provides an automatic retinal disease detection solution where many architectures of deep convolutional networks like ResNet 50, InceptionV3, and VGG-19 have been experimented with using the concept of transfer learning.A substantial performance improvement has been observed when the transfer learned deep VGG-19 model is used on a different database of OCT images as compared with the existing state-of-the-art methods.In the biomedical field, it has already been proven that deep convolutional networks pre-trained on a huge dataset performs better than the deep models trained from scratch [1]. Moreover, pre-trained deep models with transfer learning perform well and are competent enough to resolve overfitting issues in any medical image classification task [2,3]. Hence, in the proposed work, the VGG-19 model has been pre-trained on the imagenet, a huge natural image dataset. To enhance the training ability of the proposed model, transfer learning has been applied in which a few more layers have been added to the proposed model. Only the weights of convolutional layers from the pre-trained VGG-19 model have been used. The last three layers or dense layers of the proposed model have been trained using new classes so that our model is able to resolve cross-domain image categorization problems. The only purpose of pre-training transfer learning is to save time, resources, and resolve overfitting issues.

The paper is organized as follows: Section 2 includes the survey of the previous related works; Section 3 describes novel detailed research methodology and databases employed in this work; Section 4 contains the detailed results obtained during this research work; Section 5 contains a brief discussion of the employed novel approach; and lastly, the conclusion and some future directions are given in Section 6.

## 2. Related Work

For the early detection of retinal diseases, optical coherence tomography (OCT) is preferred. Srinivasan et al. [4] implemented a system based on support vector machine (SVM) and histogram of oriented gradient (HOG) descriptors. The HOG features are extracted to detect retinal diseases, specifically diabetic macular edema (DME) and age-related macular degeneration (AMD). The author used a publicly available DHU dataset of OCT scanned images and achieved a high accuracy of 100% for AMD, 100% accuracy for DME, and 86.76% accuracy for normal states of the retina. Here, the accuracy of normal is all right, but 100% accuracy of DME and AMD gives hints towards the possibility of an overfitting problem in their developed model. Alsaih et al. employed the linear SVM over their proprietary dataset to automate the classification of the OCT scanned images of DME and a normal retinal state [5]. The proposed model observed a final specificity of 87.5% and sensitivity of 87.5%. Lemaitre et al. proposed the use of exaggerated local binary patterns over OCT images data of a proprietary dataset, and it acquired overall specificity of 93.7% and sensitivity of 81.2% [6]. Later, Lu et al. developed an approach proposing the combined use of four binary classifiers as DCNN to distinguish the abnormalities from OCT scanned retinal images [7]. Their proposed approach was applied over a proprietary dataset, and their developed model attained an accuracy of 94.00% with AUC score of 0.984.

Gulshan et al. proposed a transfer learning approach over Inception architecture for the detection of DME and diabetic retinopathy from OCT fundus photography images [8]. The work achieved an accuracy of 99.1%, with good specificity of 98.1% and with good sensitivity of 90.3% over the EyePACS-1 OCT retinal images dataset. On the Messidor-2 OCT retinal images dataset, the work reported an accuracy of 99.0%, with a good specificity score of 98.1% and a good sensitivity score of 87.00%. Similarly, Karri et al. used the deep-learning based approach over the Inception model architecture in order to automate the classification process of retinal scanned images into two classes [9]. These two classes were age-related macular degeneration and diabetic macular edema. Their developed approach acquired an overall accuracy of 91.33%. The detection accuracy of age-related macular degeneration is found to be better than that of diabetic macular edema. Later, Kermany et al. used CNNs and a deep learning-based technique over Inception architecture to automate the process of classification of choroidal neovascularization, diabetic macular edema, and drusen [10]. Their proposed approach has acquired an accuracy of 96.60% and got sensitivity and specificity as of 97.8% and 97.4%, respectively. However, in their work study, the sub-dataset used in the testing process is also used in the validation process, so we can say the model is biased.

Fauw et al. used ensemble methodology—a machine learning approach in which a number of models are combined to predict the classes of the classification problem of retinal diseases [11]. Although they have achieved better performance, the complexity of model training has also increased. Rasti et al. proposed the use of an ensemblence methodology over MCME models to the given retinal diseases classification problem [12]. The researchers achieved a precision score of 98.6% and a good AUC score of 0.998. Lately Feng Li et al. also employed an ensemble methodology over four ResNet50 model architectures for automatic classification of retinal diseases [13]. The work reported an overall accuracy of 97.3%, with a good specificity score of 98.5% and good sensitivity score of 96.30%.

In [14], Burlina et al. describe an effective approach involving the use of deep learning by employing a deep convolutional neural network (Alex-Net) to automate the classification process of age-related macular degeneration and normal states of the retina from their OCT scanned images. Their developed model achieved an accuracy of 88.4–91.6% with a kappa score of 0.8. Recently, Tan et al. proposed a 14-layers deep CNN (convolutional neural network) in order to classify the age-related macular degeneration and normal states of the retina [15]. Their proposed model acquired an overall accuracy of 91.4%, with a good specificity score of 88.5% and a good sensitivity score of 92.6%. Schlegl et al. also provide a description in their work about (SRF) subretinal fluid and IRC (intra retinal cystoid fluid) in OCT scanned images, which becomes the main source in the treatment of retinal disease like AMD and DME [16]. They also used deep convolutional neural networks over their own proprietary dataset of OCT-scanned retinal images. The developed model is found to achieve an overall accuracy of 94.0% with a precision value of 91.00% and a recall value of 84.0%. The dataset majorly used in retinal disease classification are DHU dataset used in [4,9,12,17], the Mendeley OCT-Images dataset used in [10], the Bioptigen SD-OCT dataset used in [18,19], Heidelberg Spectral is HRA-OCT dataset used in [20,21], and NEH OCT-Images dataset used in [12,22]. The present research work used the Mendeley OCT-Images dataset.

In recent years, deep learning models are at the forefront of medical research. Recently, authors presented a coherent convolutional network for retinal disease detection [23]. The author made some variations in the VGG model to identify efficient features and simplify the neural network architecture. In [24], few-shot learning using generative adversarial networks has been successfully implemented to diagnose rare retina diseases. In another work, the author used a lightweight CNN named OctNET for retinal disease classification [25]. Recently, deep-learning techniques significantly contributed a lot in the diagnosis of diseases like breast cancer [26] and acute leukemia [27].

From the above literature survey, we got a direction for the present research work. We have observed that the deep learning model with transfer learning has been explored less in the medical diagnosis system. As we know, the deep learning model requires a large amount of data for good accuracy. Usually, the retinal image data provided by the clinics are not sufficient. In that situation, transfer learning can solve and prove to be more advantageous. Therefore, in the present work, we developed a transfer learning-based deep learning model for retinal diseases detection that also work for limited image dataset. We had to select appropriate model architecture and training parameters, to get a high performance detection model.

## 3. Research Methodology

### 3.1. Proposed Model

The VGG-19 model is a variant of the VGG (Visual Geometry Group) model [28]. As its name specifies the model consists of 19 layers. This pre-trained network comprises 16 convolution layers and 3 fully connected layers. It has 5 max pool layers and 1 softmax layer too. The present work proposed customization of the VGG model to achieve better results. The VGG-19 model is pre-trained only on ImageNet dataset. Therefore, customization is performed here to make our proposed model stronger in terms of identifying new images/features. The schematic illustration of the proposed work is shown in Figure 2. In the proposed model, the top-notch layers from VGG19 architecture are removed and replaced by one flatten, one dropout, and one output dense layer. Furthermore, no layers in the proposed model were frozen, and all training processes went through complete architecture. Here, “frozen” means the weights are not modified by the model after each epoch, though “frozen” enhances the training process but reduces the training accuracy. “Went through complete architecture” means weights are updated after each epoch.

Figure 3 presents the final architecture of the proposed model and is summarized as follows:Input: RGB images of size (150 × 150). The size of the input matrix is (150, 150, 3).Kernel size of 3 × 3 is used with a stride size of 1. The relationship is shown in Equations (1) and (2) [29].
(1)$$w_{nx}=w_{(n-1)x}-f_{nx}s_{nx}+1$$
(2)$$w_{ny}=w_{(n-1)y}-f_{ny}s_{ny}+1$$

Here, the size of the output feature map is represented by ${(w}_{nx},w_{ny})$, stride size is represented by ${(s}_{nx},s_{ny})$, kernel size is represented by ${(f}_{nx},f_{ny})$, and index of layers is represented by n. In the present work, the size of the input image is 150 × 150, which is difficult to show diagrammatically. Therefore, for a better understanding, a scaled-down example is taken and shown in Figure 4. The working of kernel and stride used for calculating the output feature map is shown in the diagram.

Max pooling operation is performed over 2 × 2 pixel windows with stride 2. The function of pooling layer is to reduce input dimensionality and, hence, reduce the model complexity. Max pooling operation helps in faster convergence and better generalization as it takes maximum values (information) from each sub-region [29].All layers are flattened.Finally, there is an output dense layer (None, 4), which is used for the prediction of output with softmax activation. Here, 4 represent four output classes.Softmax activation helps in multiclass classification and works on relative probabilities. Equation (3) is used for the softmax (represented by *Soft_Max*) activation function.


(3)
$$Soft\_Max(y_{i})=\frac{exp(y_{i})}{\sum_{j}{exp(y}_{j})}$$


Here, the *y* represents the values from the neurons of the output layer (i.e., output from the node). Let us take an example to understand the softmax activation function working. Consider (*y*_21_, *y*_22_, *y*_23_, and *y*_24_) as output from the output dense layer. Now the softmax activation is applied to the above output. We get the final output as follows:$$Soft\_Max(y_{21})=\frac{exp(y_{21})}{exp(y_{21})+exp(y_{22})+exp(y_{23})+exp(y_{24})}\to Prob(Class1)$$
$$Soft\_Max(y_{22})=\frac{exp(y_{22})}{exp(y_{21})+exp(y_{22})+exp(y_{23})+exp(y_{24})}\to Prob(Class2)$$
$$Soft_{Max}(y_{23})=\frac{exp(y_{23})}{\exp\left(y_{21}\right)+\exp\left(y_{22}\right)+\exp\left(y_{23}\right)+\exp\left(y_{24}\right)}\to Prob\left(Class3\right)$$
$$Soft\_Max(y_{24})=\frac{exp(y_{24})}{exp(y_{21})+exp(y_{22})+exp(y_{23})+exp(y_{24})}\to Prob(Class4)$$

The exponential acts as the non-linear function. Later, these values are divided by the sum of exponential values in order to normalize and then convert them into probabilities.Total trainable parameters in the proposed model are 20,057,156.

### 3.2. Data Used

In this research, the dataset used for analysis/training/testing is named “Large Dataset of Labeled Optical Coherence Tomography (OCT) and Chest X-ray Images” [30] and taken from the source Mendeley data where it is freely available for study and research. The dataset contributors are Daniel Kermany, Michael Goldbaum, and Kang Zhang. The dataset is arranged into three sections (training, testing, validation) and each of which is further arranged into four subsections one for each image category (normal, CNV, drusen, DME). The whole dataset consists of approx. 84 thousand OCT retinal scan images in JPEG format. These images are named as per the rule, (disease_name)-(patient_ID)-(image_number). Dataset images were collected between 1 July 2013 and 1 March 2017 from the retinal scans of adult patients at California Retinal Research Foundation, Shiley Eye Institute of the University of California, Beijing Tongren Eye Center, Medical Center Ophthalmology Associates, and the Shanghai First People’s Hospital [31].

### 3.3. Data Preprocessing

In this section, some set of key observations over the data are identified and becomes the basis of preprocessing performed over the training subset of data before actual training of the proposed deep learning model. This preprocessing and experiment design are represented clearly in Figure 5. The preprocessing process performed on the sample data comprises of following steps:(i)Synthetization of Input Image: After analyzing a set of images from all four classes (normal, drusen, DME, and CNV) it has been observed that shape of images in the training dataset is of varying size. Therefore, all the images were synthesized in the same shape of 150 × 150. This ensures our model is less error prone.(ii)Image Rescaling: Rescaling pixels of all data images into (0–1) with a rescaling factor = 1./255 on all training, validation, and test datasets.(iii)Data Augmentation: The image augmentation technique is done on the training dataset, which makes the training set much bigger and enhances the capability of the model to handle images on a different axis. The following augmentation is performed on images.

Zoom Range = (0.73, 0.9),

Horizontal Flip = True,

Brightness Range = (0.55, 0.9)

Width Shift Range = 0.10,

Rotation Range = 10,

Fill Mode = ‘constant’,

Height Shift Range = 0.10.

### 3.4. Model Training

The dataset used in this research work is already being divided into training, testing, and validation subsets. The split is done independently at the candidate level (not at data level), which simply means each and every image of a particular candidate is included in the same sub-section of the dataset (either training or testing). The architecture of the proposed VGG-19 based model is represented in Figure 3. The VGG-19 based model trained in this research work (with ImageNet pre-trained weights) has parameters, which are optimized and shaped on the validation dataset and training dataset, respectively. During the training process, a total of approx. 2600 steps with a batch size of 32 images per epoch were performed and the other parameters used in model training during this novel work are summarized in the Table 1.

### 3.5. Algorithm

The proposed approach used in this work is specialized to achieve better-automated detection of retinal diseases. This model is developed and its performance is recorded and compared to existing models. The structure shown in Figure 4 consists of layers from the pre-trained model and a few new layers. These algorithmic steps of this research work are represented visually in Figure 6.

The algorithmic workflow of this research work is described as follows:Collect data from the source [30]. Data comprises one directory, which has three sub directories, namely train, validation, and test. Data organization is explained in Figure 5.Make ImageDataGenerator, which will read images according to the directory name. Three ImageDataGenerators are made for reading images from three directories, namely train, validation, and test. Each of which further contains 4 subdirectories for DME, drusen, CNV, and normal images. Perform resize and scaling of input images to (150 × 150) as source images are of different shapes. All preprocessing is done only on a training dataset.Initialize VGG19 architecture with predefined ImageNet weights using TensorFlow, while ignoring top-notch layers and adding custom Flatten, Dropout, and Output Dense layers.Create early stopping callback function on training loss parameter and model checkpoint callback function to save only best performance weights of the proposed model.(i) Select the Bbatch size (${\left(size\right)}_{batch})$ to process images as 32 images while selecting steps-per-epoch ($S_{E}$) and validation-steps ($S_{V}$) as per the below equations:
(4)Sv=(size)train(size)batch
(5)Sv=(size)valid(size)batchHere, ${\left(size\right)}_{train}$ and ${\left(size\right)}_{valid}$ represents size of the training set and validation set, respectively.(ii) Feed the training input images in our ready model with the validation set as validation input images, and train it for 25 epochs straight.(iii) Best weights are stored automatically in local storage, as per checkpoint call-back function passed.Test the testing images dataset with the model after training and saving the best weights, and record the predicted result.Calculate the Test accuracy and Test Loss based on the predicted results, and calculate all the statistical evaluation like the confusion matrix, specificity, sensitivity, kappa, and AUC.

### 3.6. Performance Evaluation

The various performance measures have been calculated to show the effectiveness of the proposed model. In the present work accuracy, sensitivity, specificity, receiver operating characteristics (ROC) graph, area under curve (AUC), and Cohen’s kappa score has been calculated. The Equations (6)–(8) describe mathematical formulas to calculate accuracy, sensitivity, and specificity using true positive (*TP*), false positive (*FP*), true negative (*TN*), and false negative (*FN*).
(6)$$sensitivity=\frac{TP}{TP+FN}$$
(7)$$specificity=\frac{TN}{TN+FP}$$
(8)$$accuracy=\frac{TP+TN}{TP+FP+TN+FN}$$

Here, true positive (*TP*) means the count of correct classification of positive class, true negative (*TN*) is the count of correct classification of negative class, false positive (*FP*) is the count of incorrect classification of positive class, and false negative (*FN*) means the count of incorrect classification of negative class.

To visualize the performance of our model, an AUC and ROC graph has been used because of its popular use in the medical field and was computed using the methods proposed in [32]. To handle the problem of imbalanced class and multi-class problems, Cohen’s kappa (*K*) has been calculated using the Equation (9).
(9)$$K=\frac{\left({accuracy}_{\left(predicted\right)}-{accurcay}_{\left(expected\right)}\right)}{\left(1-{accuracy}_{\left(expected\right)}\right)}$$

## 4. Results

### 4.1. Evaluation of Proposed Model over Testing Data Subset

Table 2 summarizes the performance of our model around the present study and compared it with pre-existing models like ResNet50 and InceptionV3. The result of the proposed model is evaluated on an independent testing subset of the development dataset. During the multiclass comparison between the specified retinal disease classes (CNV, DME, normal, and drusen), the proposed VGG-19 network architecture with transfer learning observed an outstanding accuracy of 99.17%. The sensitivity and specificity values reported are 99.0% and 99.5%, respectively. A high AUC value of 0.9997 is also reported in the model evaluation. A comparison in computational time is also presented in the Table 2 to show that transfer learning reduces computational time.

The final classification report of the model proposed in this novel research work is shown below in Figure 7 along with the statistical evaluation parameters to support the performance of our model and approach of this study.

A confusion matrix representing specific assignment of various predicted output results for various input images of four respective states of retina is shown in Figure 8. The confusion matrix showed that the proposed model successfully classify the images into corresponding retinal diseases.

Figure 9 represents the ROC curve for the proposed model developed in this research study, which is a tradeoff between sensitivity and (1–Specificity) values of the model.

Table 3 presents some analysis of state of art techniques on the retinal disease classification problem based on deep learning models [8,9,10,14,15,16,19,22,23] and non-deep learning models [4,5,6,7,17,18,20,21].

The manual screening-based methodology can be prone to human errors [16] and has proven to be time-consuming. The proposed approach is an automated process, therefore, will provide quick, accurate, and consistent results over the retinal OCT scans. The approach described here in this novel study achieved a remarkable accuracy of 99.17% with a very high sensitivity of 99.00% and high specificity of 99.50%. The AUC was 0.9997. The VGG-19 based proposed model performed better than results obtained by [5,7,9,10,11,13,14,16,18,24], and our model shows comparable performance to those in [4,8,12,19,20,33]. The detailed description is mentioned in Table 3. As we know, OCT scans are widely used for the detection of retinal disease; regardless, doctors and ophthalmologists need a lot of experience in providing the manual classification of these diseases. Moreover, it is not feasible for ophthalmologists to inspect a large number of OCT images by themselves. The proposed transfer learning-based DL approach automates the whole process of retinal disease classification and proved to be better than the pre-existing methods (see Table 2).

### 4.2. Evaluation of Proposed Model on Public DHU Dataset

Here, in this section, we show the result of the analysis performed over a totally different and isolated dataset-DHU Dataset [4] to support the performance and accuracy of our developed model. DHU dataset is a publicly available dataset of retina OCT scans, acquired from 45 participants: normal patients (15 count), dry AMD (Age-related macular degeneration) patients (15 count), and DME patients (15 count). All these OCT images in the specified dataset were obtained by Spectralis OCT imaging (Germany) under the supervision of protocols issued by the Institutional Review Board. The imaging process was performed at Duke University, Harvard University, and the University of Michigan.

As we know, the DHU dataset contains an extra class named AMD apart from Normal and DME. The proposed model has been trained to detect the four classes of the retina- CNV, DME, drusen, and normal. Therefore, the AMD subset images have been excluded from our testing.

The performance comparison of ResNet50, InceptionV3, and the proposed VGG-19 model on the DHU dataset has been investigated as shown in Table 4. The performance of the ResNet50 model over the two classes (normal and DME) is 86.67% and 80.0%, respectively. Similarly, the performance of the InceptionV3 model over both the classes (normal and DME) is 86.67% and 86.67%, respectively. The performance of the proposed model over both the classes (normal and DME) is 100% and 100%, respectively, as shown in Table 4. The proposed model clearly outperformed the rest of the models, confirming the superiority of the proposed approach for retinal disease detection.

### 4.3. Statistical Evaluation

We implemented the proposed deep learning model and calculated the overall accuracy, sensitivity, specificity, confusion matrix, precision, f1-score, and AUC parameters for the part of statistical evaluation of our model. The ratio of the count of true positive results to the total positive data is used for calculating the sensitivity in statistical evaluation. The specificity is also evaluated as a ratio of the count of true negative results to the total negative data. The accuracy of the proposed model is calculated as a ratio of the count of true positive and true negative results to the count of total positive and total negative data. We have also plotted the receiver operating characteristics (ROC) curve, which is actually a plot between the true positive rate and the false positive rate of testing results recorded during model evaluation. The higher the AUC score value of any model, the better the model performance will be. At last, we evaluated the value; the larger the kappa value of the model, the better its reliability. These statistical evaluations were done using the Python package, sklearn library, matplotlib library, and seaborn library. These parameters are not only be used to support the performance and nature of our trained model, but also to compare the performance and validity of our work with the pre-existing models and research works, respectively.

## 5. Discussion

In the present research, a novel approach of transfer learning based VGG-19 network architecture is employed for the automatic and reliable detection of four major classes of retinal diseases, namely CNV, drusen, normal, and DME. The model is trained on a very large OCT scan images dataset of the retina, provided by Mendeley data, as 84,495 images of retinal OCT scans. An independent testing dataset (part of the development dataset) and another publicly available dataset, the DHU dataset (coming from a very different source), are used for validating the performance of the proposed model. The results of performance testing suggest that our approach has achieved a comparable accuracy with a high sensitivity and specificity with a significant AUC score for the automated detection of retinal diseases from their optical coherence tomography images of the retina. The work showed the proposed model accomplishment at a level similar to or better than the pre-existing methods of detection.

The transfer learning methodology needs a huge amount of data for model training; otherwise, there may be chances of the problem of overfitting and underfitting. To overcome these problems, the proposed model stopped the training process as soon as no improvement in the validation dataset is seen for continuous three epochs. We also have applied the data augmentation method to our image data to reduce the abovementioned problems. Regardless, there are some limitations to this work. One limitation is the way the data has been collected. We have obtained our data from Mendeley data organization where they have used the Heidelberg Spectral imaging system. The device settings and orientations could have affected the data and, in turn, may affect our model’s performance. The other limitation is the amount of data available for training and testing the model. The proposed model works when a huge amount of data is available and fails in the case of limited availability of data. DNA-based computing works wonders in such cases (limited amount of data) in comparison to deep learning [37,38,39]. DNA-based computing is nature-inspired and has been found effective in making intelligent diagnostic systems for biomedical applications [40].

## 6. Conclusions and Future Scope

In this present novel study, a deep learning based model is developed to classify the OCT scanned images of the retina into four classes of retinal diseases. The proposed model is based on the VGG-19 network architecture. The work’s motivation came from the increasing use of OCT imaging techniques and computer-aided diagnosis (CAD) in medical disease detection. We got our direction from the previous study to develop a VGG-19 architecture-based transfer learning model for retinal disease detection. This model is being trained, validated, and tested over the retinal OCT images dataset from Mendeley data using the methodology of transfer learning (with pre-trained weights of ImageNet dataset). The performance of the proposed model is not only being evaluated over the development dataset but also over the foreign dataset, the DHU dataset. The work has been compared with state-of-the-art models in retinal disease detection. A proper statistical evaluation in terms of performance, specificity, sensitivity, AUC, Cohen’s kappa value, F-1 score, etc. parameters has been carried out. The findings showed that the proposed transfer learning model works better for the automatic detection of retinal diseases in comparison to that of the state-of-the-art method. The proposed method is able to differentiate between the four states of retina—CNV, drusen, DME, and normal—with high remarkable accuracy (99.17%), sensitivity (99.00%), specificity (99.50%), and AUC (0.99917). In the future, the proposed work can be extended for the detection of other diseases related to the retina, for example, diseases like retinal tear, retinal detachment and retinitis pigmentosa, etc. The model can also be investigated on the database of OCT angiography and fundus photographs.

## Figures and Tables

**Figure 1 healthcare-11-00212-f001:**
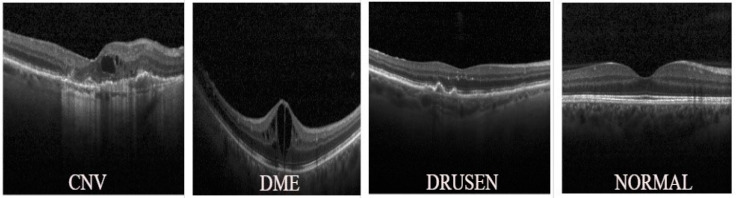
Sample OCT Images.

**Figure 2 healthcare-11-00212-f002:**
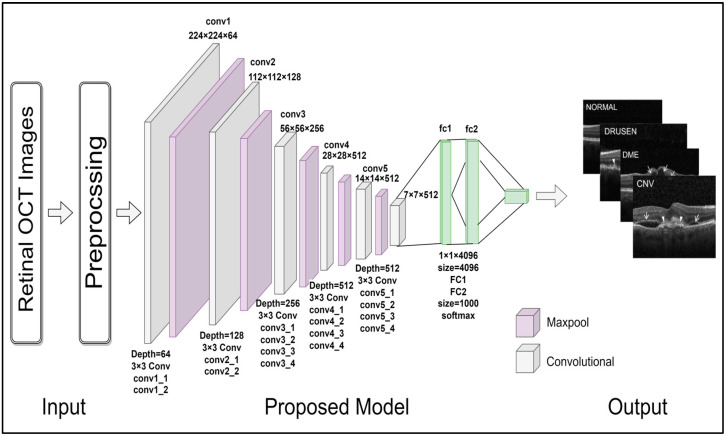
Schematic illustration of proposed work.

**Figure 3 healthcare-11-00212-f003:**
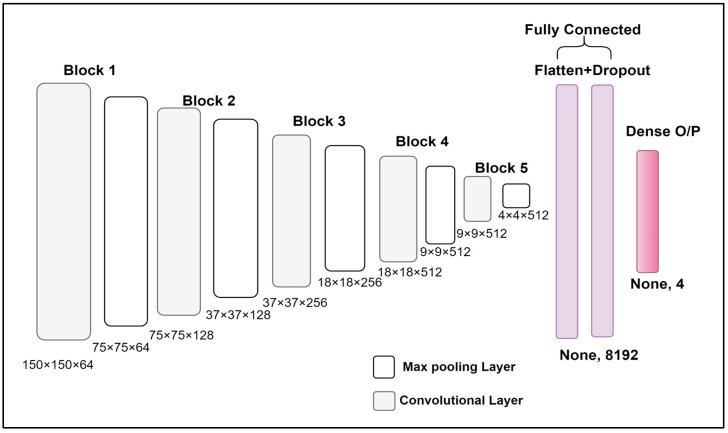
Detailed Architecture.

**Figure 4 healthcare-11-00212-f004:**
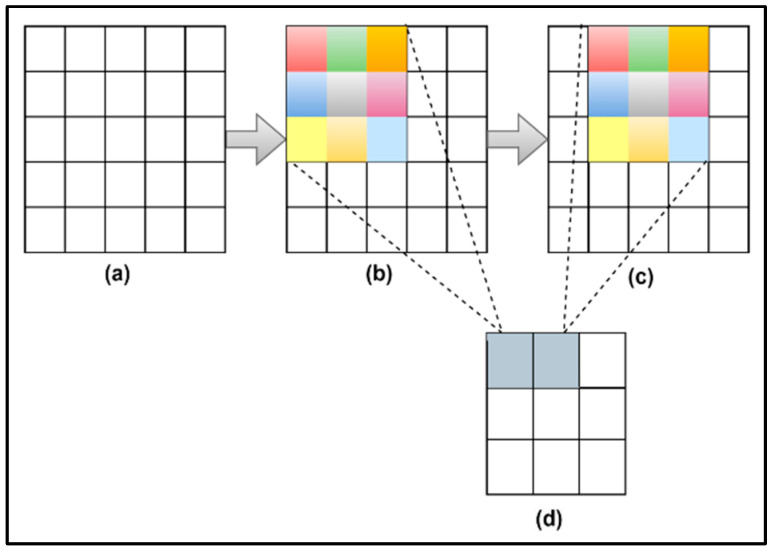
Scaled-down example (**a**) Input image of size 5 × 5 (**b**) kernel of 3 × 3 (**c**) kernel of size 3 × 3 with stride size 1 (**d**) output image of size 3 × 3.

**Figure 5 healthcare-11-00212-f005:**
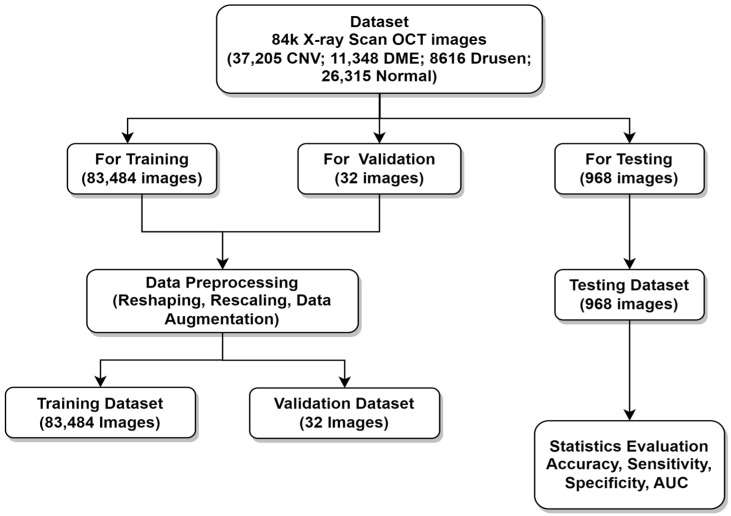
Overall experimental design used in the present research work.

**Figure 6 healthcare-11-00212-f006:**
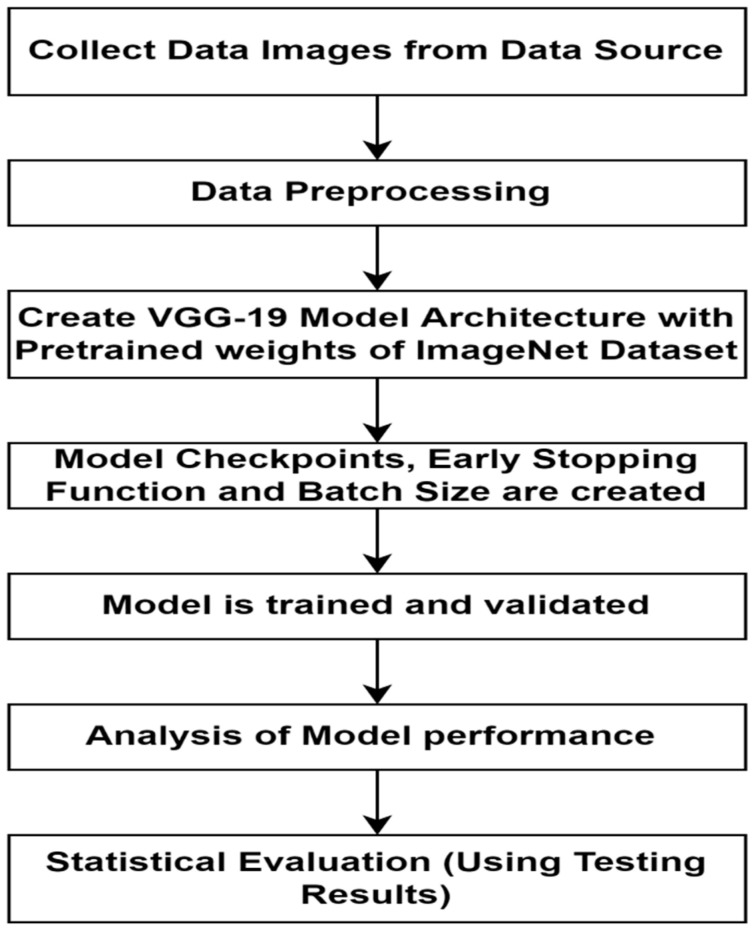
Algorithms steps for model training and evaluation.

**Figure 7 healthcare-11-00212-f007:**
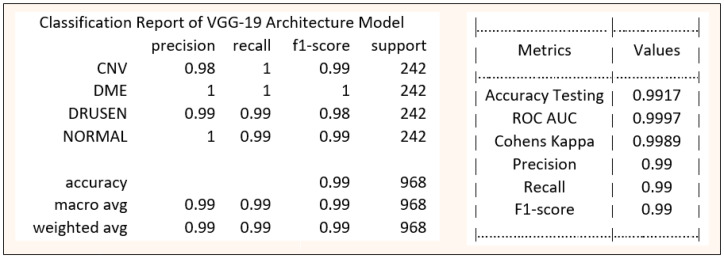
Final Classification report of VGG-19 based Model.

**Figure 8 healthcare-11-00212-f008:**
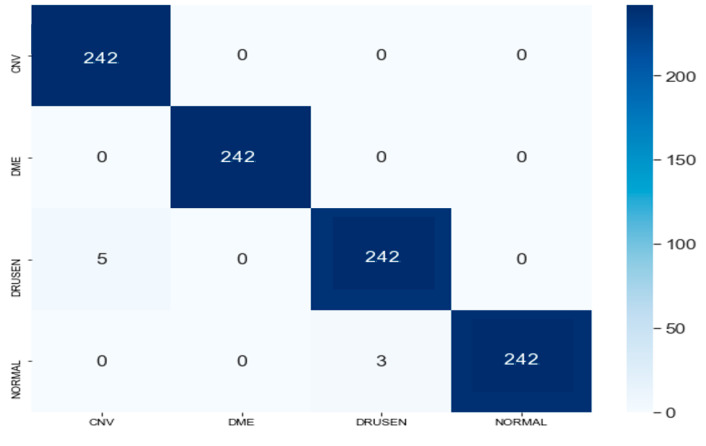
Confusion Matrix for the predictions of VGG-19 Model.

**Figure 9 healthcare-11-00212-f009:**
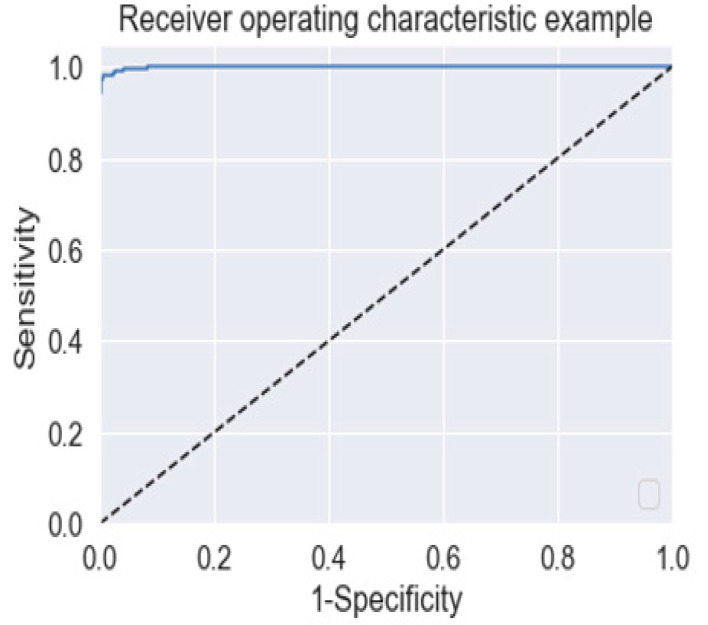
Plotted ROC Curve for model trained in this work.

**Table 1 healthcare-11-00212-t001:** Parameters and Data subsets used in proposed work.

Experiment Parameters	Specific Values
Initialized Weights	ImageNet Dataset
Dataset for Training	83,484 OCT images, divided into 4 sections
Dataset for Validation	32 OCT images, divided into 4 sections
Dataset for Testing	968 OCT images, divided into 4 sections
Input image size	150 × 150
Model Output	Softmax probability for 4 respective states of retina.
Model Training steps per epoch	Approx. 2600 steps per epoch

**Table 2 healthcare-11-00212-t002:** Performance of VGG-19 based transfer-learning model and its comparison with pre-existing models.

Parameters	ResNet50	Inception V3	Proposed Model
AUC	0.9400	0.9500	0.9997
Kappa	93.6%	94.1%	98.89%
Sensitivity	92.41%	93.60%	99.00%
Specificity	93.81%	93.90%	99.50%
Accuracy	94.61%	95.63%	99.17%
Training Time(in min)	383	393	265
Testing Time (in min)	15	18	4

**Table 3 healthcare-11-00212-t003:** Comparative analysis of retinal disease detection.

Reference	Approach	Database Used	Result and Observation
P. Srinivasan et al. [3] (2014)	Used Histogram of Oriented Gradients descriptors and SVM for Classification	Publically available DHU dataset	Accuracy for AMD = 100%, for DME = 100%, for Normal = 86.6%, (model suffered from overfitting)
Venhuizen et al. [18] (2015)	Used BOVW (Bag of Visual Words) machine-learning algorithm	Bioptigen SD OCT Dataset containing AMD and Normal retinal scans.	Overall Accuracy = 0.984
G. Lemaître et al. [6] (2016)	Implemented feature extraction using Local Binary Patterns	Proprietary	Overall Sensitivity Score = 81.20%; Overall Specificity Score = 93.70%
V. Gulshan et al. [8] (2016)	Used deep learning based 10 binary Inception V3 model architecture	Publically available (EyePACS-1 and Messidor-2) dataset	EyePACS-1 data: Overall AUC: 0.99, Overall sensitivity: 90.30%, Overall specificity: 98.1% Messidor-2 data: Overall AUC: 0.99, Overall sensitivity: 87.0%, Overall specificity: 98.5%,
Apostolopoulos et al. [19] (2016)	They have proposed to use a two-dimensional DCNN model for OCT images classification.	Bioptigen SD-OCT dataset	Achieved accuracy = 99.7%
Alsaih et al. [5] (2017)	They have employed Linear Support Vector Machine techniques for OCT classification.	Proprietary	Final Sensitivity score = 87.5% and Specificity score = 87.5%
Karri et al. [9] (2017)	Inception network	Publicly available DHU dataset of OCT scanned images of 45 patients.	Achieved accuracy for AMD = 99.0%, for DME = 89.0%, for Normal = 86.0%
Venhuizen et al. [20] (2017)	They have proposed to employ Bag-of-Visual-Words algorithm for features extraction, for detection of AMD.	Heidelberg Spectralis HRA-OCT dataset of 3265 eyes retinal images.	Accuracy = 98.0%, Sensitivity = 98.2%, Specificity of = 91.2%.
Sun et al. [21] (2017)	They have developed an automated retinal disease detection model, applying partitioning of image with feature extraction using SIFT over the Linear SVM model.	Heidelberg Spectralis HRA-OCT dataset of 3265 eyes retinal images.	For AMD, DME and Normal states of retinal their work have achieved a Cognitive Ratio of 100%, 100%, 93.3% on first set testing images and on second set of testing images they achieved Cognitive Ratio of 99.67%, 99.67%, 100% respectively.
Burlina et al. [14] (2017)	Deep Learning DCNN-AlexNet	National Institute of Health AREDS	Accuracy: (88.4–91.6)%, kappa score: 0.8
Hussain et al. [17] (2018)	Random Forest algorithm	DHU dataset and a proprietary dataset.	Accuracy of 94.0%, Final AUC score of 0.990,
Tan et al. [15] (2018)	14-layer deep CNN	Public (Kasturba Medical College)	Overall Accuracy 95.47%, sensitivity score = 96.4%, specificity score = 93.7%
Lu et al. [7] (2018)	Combination of 4 binary classifiers networks for detection of retinal diseases.	Own proprietary OCT scanned dataset.	The work has achieved an accuracy of 95.90%. Sensitivity is observed as 94.00% with specificity of 97.30% over 10-fold cross validation.
D. Kermany et al. [10] (2018)	Inception V3 architecture	Mendeley dataset of OCT scanned images.	Achieved overall accuracy of 96.60% with specificity and sensitivity of 97.40%, 97.80% respectively,
J. De Fauw et al. [11] (2018)	ensemble methodology over two model networks as DenseNet and U-net	OCT scanned dataset of retinal images	Overall Accuracy of 94.50% with AUC score of 0.992 over first set of testing images; Overall Accuracy of 96.60% with AUC score of 0.999 over second set of testing images,
Rasti et al. [12] (2018)	MCME model with ensemble methodology.	DHU dataset, NEH dataset of retinal OCT images.	Over DHU images dataset: Precision of 98.3%, with Recall of 97.78% and AUC score of 0.99 Over NEH images dataset: Precision of 99.3%, with Recall of 99.3% and AUC score of 0.9980
Vahadane et al. [13] (2018)	Patch based Deep Learning	Proprietary	Precision: 96.43% Recall: 89.45%
Schlegl et al. [16] (2018)	Deep Learning DCNN	Proprietary	Accuracy: 94.00% Precision: 91.00% Recall: 84.00%
F. Li et al. [22] (2019)	Transfer Learning over VGG-16 network	Proprietary	The proposed work achieved overall Accuracy of 98.60%, Specificity of 99.4%. and Sensitivity of 97.8%
L. Fang et al. [33] (2019)	LACNN based model.	NEH OCT images dataset, UCSD OCT images dataset.	Over NEH images dataset: Sensitivity of 99.3% with precision of 99.39% and AUC score of 0.994. Over UCSD images dataset: Accuracy of 90.10% with precision and sensitivity of 86.20% and 86.80% respectively.
Feng. Li et al. [34] (2019)	Employed ensemble methodology over four models each based on the ResNet50 network.	Proprietary	Achieved best performance as accuracy of 97.90%, with Specificity: 99.4% and Sensitivity of 96.80%. Also Kappa value of 0.9690, AUC score of 0.9980.
Shankar K et al. [35] (2020)	deep learning model named as synergic deep learning (SDL)	Messidor DR dataset	Accuracy = 99.28, Sensitivity = 98.54, Specificity = 99.38
Vives-Boix et al. [36] (2021)	CNN (Am-Inception V3)	APTOS (Asian Pacific Tele-Ophthalmology Society) 2019	Accuracy = 94.46%
**Proposed work**	**Transfer learning based deep learning model**	**Mendeley dataset of OCT-scanned images**	**Accuracy = 99.17%, sensitivity = 99.00% and high specificity of 99.50%.**

**Table 4 healthcare-11-00212-t004:** Comparison with other models using DHU dataset.

Model Architecture	% of Volume Correctly Identified	
	Disease Class	
	Normal (Out of 15)	DME(Out of 15)
ResNet50 Model	86.67%	80.0%
InceptionV3 Model	86.67%	86.67%
**Proposed Model**	100%	100%

## Data Availability

We have utilized the following two public datasets in this study: Ref [4] available at https://people.duke.edu/~sf59/Srinivasan_BOE_2014_dataset.htm (accessed on 20 November 2022) and ref [32] available at 10.17632/rscbjbr9sj.3.

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
