# Peer review of "A Deep Learning-Based Framework for Retinal Disease Classification"

_healthcare, 2023, doi:10.3390/healthcare11020212_

Round 1

Reviewer 1 Report (New Reviewer)

The authors present a well-known convolutional neural network (CNN) model applied to retinal disease detection. The experimental results are reported.

The authors used a transfer learning on the image data sets to solve a specific problem of medical imaging, related to the retinal disease. The authors have used a relatively large data set of retina disease images. 3 CNN models were tested, and the authors observed that the less complex model VGG-19 outperforms more advanced ResNet50 and Inception3 models. Data partition schema is presented, as well as the detailed training parameters description. 

The authors use only one data set, that makes us recommend the use of other data sets. For example, data sets of a smaller size, that can be common in medical imaging.. The authors split the data set into small subsets that makes the generalization task trivial. In this context, the authors are recommended to include different sizes and data partitions. The proposed method is a plain modification of the VGG-19 model provided with the well-known description. Claims that one model is better than another are generally expected to be supported with confidence intervals or hypothesis tests. More models could be compared, in the medical domain such as UNet. New generation of models such as visual transformers would be also useful to compare.

The authors are recommended to update the references in part of engineered features 

Exploring problems of the transfer learning, the authors are expected to address the above issues and in summary to use more data sets, non-trivial data set partition, more original/novel modification of ANNs and wider architecture coverage, hypothesis tests and more rigorous comparison with other models.

Author Response

Reviewer 1

The authors present a well-known convolutional neural network (CNN) model applied to retinal disease detection. The experimental results are reported.

 The authors used a transfer learning on the image data sets to solve a specific problem of medical imaging, related to the retinal disease. The authors have used a relatively large data set of retina disease images. 3 CNN models were tested, and the authors observed that the less complex model VGG-19 outperforms more advanced ResNet50 and Inception3 models. Data partition schema is presented, as well as the detailed training parameters description. 

Comment #1

The authors use only one data set, that makes us recommend the use of other data sets. For example, data sets of a smaller size, that can be common in medical imaging.

Author Response

As suggested, we have added another data set in section 4.2.

Comment #2

The authors split the data set into small subsets that makes the generalization task trivial. In this context, the authors are recommended to include different sizes and data partitions.

Author Response

Thanks for the incisive comment. Our aim is to make an automatic retinal disease detection solution where many architectures of Deep Convolutional Networks like ResNet 50, InceptionV3, and VGG-19 have been experimented with using the concept of transfer learning.  The experimental results of related models on the dataset publicly available have been compared to facilitate the research of people in related fields. We may subsequently include research on related recommendations in our future research.

Comment #3

The proposed method is a plain modification of the VGG-19 model provided with the well-known description. Claims that one model is better than another are generally expected to be supported with confidence intervals or hypothesis tests. More models could be compared, in the medical domain such as UNet. New generation of models such as visual transformers would be also useful to compare.

Author Response

We deeply appreciate your constructive comments. It would have been interesting to explore the new generation models. In our work, this would not be possible because we just focused on pre-trained deep models with transfer learning. We compared the performance with other models. However, we are thinking of including a comparison with UNet and with other visual transformers in future research work.

Comment #3

The authors are recommended to update the references in part of engineered features.

 Author Response

 Thanks for the comment. In the present work, we have used a pre-trained VGG-19 model which has been trained on the ImageNet dataset. Here, the feature extraction process is performed automatically by the pre-trained model itself. So for this work, we don’t need to do feature engineering manually.

Reviewer 2 Report (New Reviewer)

Different Evaluation parameters can be used to test the results, Transfer learning model can be changed to check the performance of algorithm.

Suggestion for Authors:

1. Objectives and hypothesis are not clear.
2. Write problem Statement clearly.
3. Comparative literature survey, will show case your research work 
4. In Introduction part add few more line about proposed work which will give clear idea about your research.
5. Evaluation parameters are not sufficient to prove the clear results add cross fold validation techniques if possible.
6. Future scope and merit of research work authors needs to add.

Author Response

Reviewer 2

Different Evaluation parameters can be used to test the results, Transfer learning model can be changed to check the performance of algorithm.

Suggestion for Authors:

Comment# 1 Objectives and hypothesis are not clear.

Author Response  Thanks for the suggestion. We have updated the objective of the present research in the manuscript in lines 64 to 75.

Comment# 2 Write problem Statement clearly.
Author Response Thanks for the suggestion. We defined the problem statement in section 1 in lines 48-69.

Comment# 3 Comparative literature survey, will show case your research work 
Author Response Thanks for the suggestion. We have updated table 3 in the manuscript.

Comment# 4 In Introduction part add few more line about proposed work which will give clear idea about your research

Author Response

Thanks for the comments, we have added one para (75-93) in the paper.

Comment# 5 Evaluation parameters are not sufficient to prove the clear results add cross-fold validation techniques if possible.

Author Response Thanks for suggesting a cross-folding validation technique, which we are planning to include in our future work.

Comment# 6 Future scope and merit of research work authors needs to add.

Author Response As suggested, we have added future scope in the last line of section 6.

Reviewer 3 Report (New Reviewer)

The contributions of the paper are OK. However, authors are advised to do the following modifications.

1.      Highlight your contributions more precisely.

2.      Highlight the limitations of existing works discussed in the literature (related work).

3.      Add one paragraph and briefly discuss more recent works including the following deep learning-based works [1-11] that have been employed in medical image processing to strengthen your literature section.  

[1]   “Coherent convolution neural network based retinal disease detection using optical coherence tomographic images,” Journal of King Saud University-Computer and Information Sciences, 2022.

[2]   “Feasibility study to improve deep learning in OCT diagnosis of rare retinal diseases with few-shot classification,” Medical & biological engineering & computing, vol.  59, pp.  401-415, 2021.

[3]   An Efficient Detection and Classification of Acute Leukemia using Transfer Learning and Orthogonal Softmax Layer-based Model”, IEEE/ACM Transactions on Computational Biology and Bioinformatics, vol. 1, pp. 1-12, 2022.

[4]   “Octnet: A lightweight cnn for retinal disease classification from optical coherence tomography images,” Computer methods and programs in biomedicine, vol. 200 pp.  105877, 2022.

[5]   High Accuracy Hybrid CNN Classifiers for Breast Cancer Detection using Mammogram and Ultrasound Datasets,” Biomedical Signal Processing and Control, vol. 80, pp. 104292, 2023.

4.      Improve the performance analysis by comparing the proposed method with other recent deep learning-based or transfer learning-based methods.

5.      Give performance analysis in terms of computational time.

6.      Discuss some weaknesses of the proposed method.

Author Response

Reviewer 3

The contributions of the paper are OK. However, authors are advised to do the following modifications:

Comment# 1  Highlight your contributions more precisely.

Author Response

As suggested by the reviewer, we have highlighted the contribution in lines 75 to 92.

Comment# 2  Highlight the limitations of existing works discussed in the literature (related work). 

Author Response

As suggested by the reviewer, we have highlighted the limitations of existing research in lines 171-176.

Comment# 3   Add one paragraph and briefly discuss more recent works including the following deep learning-based works [1-11] that have been employed in medical image processing to strengthen your literature section.  

[1]   “Coherent convolution neural network based retinal disease detection using optical coherence tomographic images,” Journal of King Saud University-Computer and Information Sciences, 2022.

[2]   “Feasibility study to improve deep learning in OCT diagnosis of rare retinal diseases with few-shot classification,” Medical & biological engineering & computing, vol.  59, pp.  401-415, 2021.

[3]   “An Efficient Detection and Classification of Acute Leukemia using Transfer Learning and Orthogonal Softmax Layer-based Model”, IEEE/ACM Transactions on Computational Biology and Bioinformatics, vol. 1, pp. 1-12, 2022.

[4]   “Octnet: A lightweight cnn for retinal disease classification from optical coherence tomography images,” Computer methods and programs in biomedicine, vol. 200 pp.  105877, 2022.

[5]   “High Accuracy Hybrid CNN Classifiers for Breast Cancer Detection using Mammogram and Ultrasound Datasets,” Biomedical Signal Processing and Control, vol. 80, pp. 104292, 2023.

Author Response

We agree with the reviewer’s assessment. Accordingly, we have included one para on recent work in medical image processing. Please see lines 163 to 170.

Comment# 4 Improve the performance analysis by comparing the proposed method with other recent deep learning-based or transfer learning-based methods.

Author Response

Thanks for the comment. We have compared the proposed method with other deep learning models and mentioned the performance in Table 2.

Comment# 5. Give performance analysis in terms of computational time.

Author Response

Thanks for the incisive comment. We have updated the analysis process and updated table 2 accordingly.

Comment# 6 Discuss some weaknesses of the proposed method.

Author Response

As suggested, we have include the weakness of proposed model in lines 459- 465.

We deeply appreciate your constructive comments.

Round 2

Reviewer 1 Report (New Reviewer)

The authors do want to provide the responses to critical suggestions on the experimental results. The novelty is also a critical area which is expected to be improved.

This manuscript is a resubmission of an earlier submission. The following is a list of the peer review reports and author responses from that submission.

Round 1

Reviewer 1 Report

The work is now acceptable in current form.

Author Response

Thank you so much for your comment.  

Reviewer 2 Report

The work needs redo. Hints are as follows:

Please take out all abbreviations from abstract. Abstract is for general readers. Abbreviations may not make sense for general readers. Replace “etc.” by “alike.”

The referece numbering does not go along with the journal polocity.

Please give an arbritary example showing what is happed to an image if equations 1, 2, and 3 are applied.

Please be consistent: (150by150) or (150px by 150px), 2*2 or 2 by 2.

Softmax in the text and softmax in the equation are different; please be consistent.

Figures 2 and 3: fonts are not visible; please increase the fonts. Please avoid colors.

Figure 4: please remove color and make it black and white.

Lines 277 and 278: please write the equation properly (number is and symbolize)

Figure 5: the text says 10 steps but figure 5 is given by 7 steps. Please be consistent. Please remove color.

Abbreviations: there are lots of abbreviations: CNV, DME, NORMAL, DRUSEN, VGG-19, AUC

Evaluation parameters: accuracy, kappa, AUC, and alike. These parameters are not defined. Please define using an appendix or put them in the methodology section (section 3).

Last paragraph of Discussion. It is very brief. Please elaborate. For example, when window size and datasets are very small and limited, respectively, DNA-based computing works well; better than deep learning. This issue can be discussed here. (DNA based computing: https://doi.org/10.3390/ma14081899, https://doi.org/10.1016/j.cirpj.2011.02.002, https://doi.org/10.1016/j.biosystems.2014.01.003).

Round 2

Reviewer 2 Report

The authors have improved the work. Still it needs redo. The abstract is still not convey the message. The reviewer has rewrite it as follows.

 <This study addresses the problem of automatic detection of disease states of retina. In order to solve the abovementioned problem, this study develops an artificially intelligent model. The model is based on a customized 19-layer deep convolutional neural network called VGG-19 architecture. The model (VGG-19 architecture) is empowered by transfer learning. The model is designed so that it can learn from a large set of images taken by Optical Coherence Tomography (OCT) and classify them into four conditions of the retina: 1) Choroidal Neovascularization, 2) Drusen, 3) Diabetic Macular Edema, and 4) Normal Form. The training datasets (taken from publiclly available sources) consist of 84568 instances of OCT retinal images. The datasets exhibit all four classes of retinal disease mentioned above. The proposed model achieved a 99.17% classification accuracy with 0.995 specificities and 0.99 sensitivity, making it better than the existing models. In addition, the proper statistical evaluation is done on the predictions using such performance measures as 1) area under the receiver operating characteristic curve, 2) Cohen’s kappa parameter, and 3) confusion matrix. Experimental results show that the proposed VGG-19 architecture coupled with transfer learning is an effective technique for automatically detecting the disease state of a retina.>

The authors must understand that "X-RAY OCT" does to make sense. It should have been OCT only.

This way the work still needs major revision. Some of the gaps are stated below.

The new figure in Section 3 does not convey any message. Please give an example using one of the datasets and show how it is preprocessed, and how the patterns are detected using VGG-19. Now we do not see such detail.

Please make it clear why you need transfer learning. Now transfer learning remains a blackbox in the manuscript.

The definitions (equations) have very unusal symbols. Some of the symbols are still not mentioned in the text. What is yi, why not it is yij?...

The authors in the discussions mentioned about two types of DNA based computing (in silico and in vitro). The reviewer does not see any connection. Please elaborate.
